# Approaches and Applications of *Mentha* Species in Sustainable Agriculture

Kalliopi I. Kadoglidou *[ID] and Paschalina Chatzopoulou

Institute of Plant Breeding and Genetic Resources, Hellenic Agricultural Organization–Dimitra (ELGO-Dimitra), Thermi, GR-570 01 Thessaloniki, Greece
* Correspondence: pkadoglidou@elgo.gr

**Abstract:** The potential applications of the genus *Mentha* as natural pesticides and environmentally friendly postharvest treatments for agricultural products in sustainable agriculture are reviewed here. The non-commercialized *Mentha* species in Greece and the rest of the world are presented, and the possibility of their exploitation is discussed. Recent developments in successive methods of application—i.e., the formulation of *Mentha* extracts/oils in eco-friendly pesticides, fumigation, and the assessment of performance in field trials—are also reported. Several studies have shown that *Mentha* species in various forms (e.g., essential oil (EO), aqueous extract, fresh or dried plant material, compost, etc.), or in different cropping systems (e.g., crop rotation, intercropping system/cover crop, cultivation and incorporation as green manure) offer the potential to be used in agriculture, with the goal of managing plant pathogens (bacteria and fungi), animal pests (insects, acarines, and nematodes), weeds, and for the improvement of soil quality and productivity as well. Finally, several studies are presented concerning the enhancement of *Mentha* EOs effectiveness in agriculture, and to also overcome the limitations of their nature (being unstable when exposed to light and oxygen), by using a combination of EOs, or by developing novel formulations (microencapsulation). Taking into consideration all the abovementioned findings, it is evident that *Mentha* species could be used in sustainable agricultural systems for integrated pest management. This can be achieved utilizing them either directly as a crop, green manure, or compost, or indirectly by developing natural pesticides based on their EOs or extracts. Nonetheless, further field experiments must be conducted, to confirm the efficacy of various formulas on pests under crop conditions.

**Keywords:** extracts; essential oils; microorganisms; insects; acarines; nematodes; weeds; crops; phytotoxicity

## 1. Introduction

The genus *Mentha* is a taxon in the Lamiaceae family, which includes 18 species and 11 hybrids that grow globally, especially in South Africa, Australia, and in mild climate regions of Eurasia [1]. *Mentha* is classified in the tribe Mentheae, and the systematics of the section *Mentha* are particularly complicated, due to the easy hybridization within species and the existing polymorphism. The *Mentha* genus is divided into four sections: *Pulegium*, *Tubulosae*, *Eriodontes*, and *Mentha*; though eleven natural hybrids have been developed from the species *M. arvensis*, *M. aquatica*, *M. spicata*, *M. longifolia*, and *M. suaveolens* [2]. Today, the highest diversity occurs in Western Europe. However, endemic species have been found in other continents, such as in Eastern and Western North America, Asia, South Australia, and Tasmania. In addition, several naturalized, introduced, or cultivated *Mentha* species are grown in numerous districts all over the world.

According to Dorman et al. [1] the most important mint species commercially are spearmint (*Mentha viridis* L., syn. *M. spicata*), mint or peppermint (*M. × piperita*), and corn mint (*M. arvensis*, syn. *M. canadensis*), mostly due to their essential oils (EOs), which are

high traded worldwide, the annual production of which is more than 23,000 metric tons and outrun USD 400 million/year [3].

Spearmint and mint originate from the Mediterranean area, where they have been found as native populations. For example, in the case of Greece, Karousou et al. [4] referred to the *M. longifolia* subsp. *petiolata* and *M. × villoso-nervata* as two non-commercialized wild *Mentha* species grown in Crete. Similarly, Kokkini et al. [5] referred to ten wild *M. pulegium* populations growing in Crete that were varied in their essential oil composition, possibly due to different climate and ecological conditions. Three other *Mentha* species growing wild in Greece are mentioned in a comprehensive review from Tucker and Naczi [2]: *M. longifolia* subsp. *erminea*, *M. longifolia* subsp. *grisella*, and *M. spicata* subsp. *condensata* (type Laconia).

Spearmint and mint include many varieties, while mint is the sterile bispecific hybrid *M. aquatica × M. spicata.* Mint was cultivated first in the Mediterranean basin, whereas its commercial cultivation in England began at the late 1700s. Mint was transferred to America from Europe at the same time [6]. Both mint and spearmint are perennial species [7] and produce stolons, which are thin rhizomes that grow either underground or aboveground [2].

*Mentha* sp., are characterized by their distinct flavor, and several species have been used for centuries as condiments, in tea preparations, and for medicinal purposes [3]. Recently, Vining et al. [8] reviewed the species distribution worldwide, its uses through centuries, its domestication history, and itsbreeding aspects. All *Mentha* species constitute EO plants, thus, being significant species of the Lamiaceae, due to their high economic value [9]. Both mint and spearmint are aromatic and medicinal plants, utilized fresh or dried as condiments, in cooking, herbal teas, etc. [10], while EOs are used in the food and drink sector, i.e., confectionary, beverages, bakery, in pharmaceuticals and hygiene products, in perfumery, cosmetics, pesticides etc. [7]. Moreover, the biological activities of the *Mentha* species, i.e., antioxidant, antibacterial, antiviral, antifungal, etc., have been extensively studied.

The interest in the exploitation of several *Mentha* species as biopesticides in the context of organic farming and food production is increasing. In fact, among the *Mentha* species growing worldwide, the most studied as biopesticides are *M. × piperita* (peppermint), *M. spicata*-syn. *M. viridis* (spearmint), *M. pulegium* (pennyroyal), *M. longifolia* (wild mint or horsemint), *M. arvensis* (corn mint or wild mint), *M. suaveolens* (apple mint), and *M. rotundifolia*. A co-occurrence analysis recently conducted by Catani et al. [11] revealed that *Mentha* EOs are included among the eight more frequently used EOs in agriculture, mostly as alternative plant protection products. The EOs, as well as other bioactive compounds of the *Mentha* genus, exhibit a broad spectrum of actions as biocides in agriculture, affecting microorganisms (like bacteria, fungi, and yeasts), animals (like insects, acarines, and nematodes), and plants (like weeds and crops). The high effectiveness of their EOs is due to their main compounds, e.g., menthol, menthone, pulegone, carvone, 1,8-cineole, limonene, and b-caryophyllene. Mint and peppermint oils derived from *Mentha* spp. and *M. × piperita*, respectively, with both menthol and menthone as major constituents, are among the most common EOs used in pesticide formulation [12]. The variable activities of the different *Mentha* species are associated with different EOs chemotypes that sometimes occur within the same species. The allelopathic properties/efficacy of these EOs have been reported in several studies, whereas the majority are generally referred to in vitro bioassays [13].

The comparison of different results concerning the biocide activities of *Mentha* EOs reported by different researchers is difficult because of the effect of numerous variables. A problem when comparing the effectiveness of EOs, reported in the literature, is the lack of information regarding the concentration used. Other difficulties result from the various applied techniques (e.g., disk diffusion, agar or broth dilution methods), different protocols (e.g., assessing the Minimal Inhibitory Concentration (MIC), evaluation of the survival curves, electron microscopy scanning analysis), and various different metric systems as they are reported in relevant studies. In particular, significant inconsistencies arise when

comparing the bioactivity and effectiveness of a *Mentha* species, both under in vitro and in vivo conditions.

In this work, the efficacy of EOs, extracts, and various plant material/tissues belonging to the *Mentha* genus against organisms harmful for the agriculture are presented in the context of integrated pest management and the principles of sustainable agriculture. Additionally, the mode of action, the current and the prospective tendencies/challenges, like the utilization of *Mentha* species in the crop rotation system, or as soil amendment, as well as their use in novel formulations, are also discussed.

## 2. Target: Microorganisms

### 2.1. Antibacterial Activity

Singh and Pandey [14], suggested, in their review on antibacterial, antifungal, and insecticidal activities of *Mentha* EOs, that they are promising as natural pesticides against plant microbial pathogens or storage insect pests, with commercial value. Bacteria and fungi cause 40–50% loss of agricultural production, so they are also among the most important pathogens of crops and stored food commodities [15]. According to Vidhyasekaran [16], serious damage caused by bacteria throughout the world are estimated to cause losses of 30–40% per year in crops, as well as in the postharvest stage. The main genera causing the most destructive injuries/diseases are *Xanthomonas*, *Pseudomonas*, and *Erwinia* [17]. Finding a solution to the problem is becoming more and more imperative, since the plant pathogenic bacteria develop resistance to copper bactericides and streptomycin [18]. The effect of *Mentha* EOs on plant pathogenic bacteria employing in vitro techniques like agar dilution, disk diffusion, and broth dilution methods has been studied [19]. In a more detailed study, Işcan et al. [20] found that *M. × piperita* EOs exhibited antibacterial activity by broth dilution bioassay, ranging from 0.07 to 1.25 mg/mL of the minimum inhibitory concentration (MIC) values, against the following plant pathogenic bacteria: *P. syringae* pv. *syringae*, *Pseudomonas syringae* pv. *tomato*, *P. syringae* pv. *phaseolicola*, *Xanthomonas campestris* pv. *campestris*, and *X. campestris* pv. *phaseoli*. The same trend was observed for the major EO compounds menthol and −(-)menthone. Moreover, it was found that EOs constituents like menthol, neomenthol, isopulegone, and 1,8-cineole, at 20 μL in a disc diffusion assay, significantly inhibited the growth of *Acidovorax citrulli* bacterium induced fruit blotch in watermelon in vitro [21]. Additionally, at a concentration of 0.2%, the same constituents, as well as the peppermint EO, prevented bacterial growth in vivo, whereas at 0.1%, menthol and neomenthol resulted in 50% inhibition in bacterial growth, though isopulegone resulted in 83% inhibition, and 1,8-cineol and peppermint oil resulted in 92% inhibition, respectively. Different genera of bacteria respond differently to *Mentha* EOs, according to Vasinauskienë et al. [22], who found, by using a disk diffusion assay, that EO of *M. × piperita* exhibits a strong inhibitory effect (6–12 mm zone of inhibition) against *Xanthomonas vesicatoria*, and moderate inhibition (2–6 mm zone of inhibition) against *Erwinia carotovora* subsp. *carotovora*, *Bacillus* sp. and *Pseudomonas syringae* pv. *tomato*, respectively. However, no effect was observed on *Pseudomonas marginalis* pv. *marginalis* and *Pseudomonas syringae* pv. *syringaea*. Moreover, Shetta et al. [23] found that, encapsulated in chitosan nanoparticles, *M. × piperita* oil showed an enhanced antibacterial potency against *Staphylococcus aureus* (Minimum Bactericidal Concentration, MBC, 0.57 mg/mL), whereas the pure *M. × piperita* oil showed more powerful antimicrobial properties than the nanoencapsulated one against *Escerichia coli* (MBC 1.15 mg/mL).

Soltani and Aliabadi [24] investigated the antibacterial activity of aqueous extracts and EOs of *M. spicata* and *M. × piperita* against *Xanthomonas arboricola* pv. *juglandis* that causes the bacterial blight of walnut, the most destructive bacterial disease of the genus *Juglans* worldwide. They observed that the application of both extracts and EOs of *M. × piperita* and *M. spicata*, by using diffusion assays, showed in vitro the highest antibacterial activities against *Xanthomonas arboricola* pv. *juglandis*.

Concerning *M. pulegium*, reports show contradictory results for its efficacy against bacteria. Studies showed that *M. pulegium* exhibited a broader antibacterial activity than other

EOs; it was more effective with lower minimum inhibitory and/or lethal concentration, while it presented equal or stronger antibacterial activity than known antibiotics, such as gentamycin, chlorophenicol, erythromycin [25,26]. Contrariwise, El Asbahani et al. [27] stated that *M. pulegium* EO was less drastic against bacteria, fungi, and yeast than other EOs. On the whole, *M. pulegium* EO can control a broad range of target microorganisms, but in some specific cases may be not the most effective. El Asbahani et al. [27] found that *M. pulegium* EO was more efficient against gram-negative bacteria than gram-positive ones. Contrary to this, Sarac and Ugur [28] referred that bacteria belonging to the *Pseudomonas* genus were unaffected by *M. pulegium* EO. Another property of *M. pulegium* oil is that even at concentrations below MIC, it can cause the dispersal of the bacteria's biofilm (a defense mechanism of bacteria against antimicrobial agents), making it susceptible to bactericides [29].

Kokoskova et al. [30] found that *M. arvensis* was effective against *Erwinia amylovora* and *P. syringae.* pv. *syringae*, as the antimicrobial efficacy index was rather almost 20%. *E. amylovora* is the most significant bacterial disease of apple, pear, hawthorn, cotoneaster, and other members of Rosaceae family [31]. *P. syringae* has seriously affected many crop and orchard industries with its various strains, causing a variety of symptoms, i.e., blossom blast, spur dieback, leaf necroses, bark cankers, and gummosis of woody tissue, and significant losses to stone fruits [32]. Considering the importance of the aforementioned diseases in agriculture, it is evident that *M. arvensis*'s antibacterial efficacy has value for the management of those diseases. Kokoskova et al. [30] declared that in the in vitro experiment with agar plates, *M. arvensis* oil exhibited up to 50% higher efficiency than streptomycin (used as a standard) against both *E. amylovora* and *P. syringae.* Pv. *Syringae*, since 1 µL *M. arvensis* oil was more drastic than 0.02% streptomycin. The in vitro antibacterial activity of *M. suaveolens* was examined against *Pseudomonas savastanoi* pv. *Savastanoi* and *Clavibacter michiganensis* subsp. *michiganensis*, indicating that *Mentha* EO was drastic only against *C. michiganensis*, with an MIC of 0.78 mg/mL [33].

In summary, in the abovementioned studies the range of variability observed in the effectiveness of different *Mentha* species against the same bacteria species is possibly explained by the different experimental methodology, the variance in the chemical components of EOs of different *Mentha* species or chemotypes, as well as the variability in bacterial strains used.

Concerning the active compounds of *Mentha* EOs, menthol is classified among the eight most effective constituents of 21 tested oxygenated monoterpenes of various EOs against 10 gram-positive and 20 gram-negative strains. Several bacteria species were sensitive to menthol, like *Aerococcus viridans*, *Clavibacter michiganense*, *Kocuria varians*, *P. syringae* pathovars, two of four *Erwinia* spp., three *Xanthomonas* taxa, *Neisseria subflava*, and *Agrobacterium tumefaciens* [34]. In contrast with menthol that hindered the growth of 16 strains, menthone inhibited the growth of only two strains.

The antibacterial activity of *Mentha* species (EOs, extracts, etc.) against phytopagonenic bacteria is summarized in the Table 1.

**Table 1.** Antibacterial activity of *Mentha* species (EOs, extracts, etc.) against phytopathogenic bacteria in the cited literature.

| Target Bacterium | Species of *Mentha* | Tested Compound(s) | Activity/Toxicity | Method(s)/Dose | Reference |
|---|---|---|---|---|---|
| *Pseudomonas syringae* pv. *syringae*, | *M.* × *piperita* | EO [1]/menthol/menthone | MIC [2]: 0.07–1.25/0.156/2.5 mg/mL, respectively | Broth dilution bioassay | [20] |
| *P. syringae* pv. *tomato*, | | EO/menthol/menthone | MIC: 0.07–1.25/0.07/1.25 mg/mL, respectively | | |
| *P. syringae* pv. *phaseolicola*, | | EO/menthol/menthone | MIC: 0.07–1.25/1.25/2.5 mg/mL, respectively | | |
| *Xanthomonas campestris* pv. *campestris*, | | EO/menthol/menthone | MIC: 0.07–1.25/0.156/1.25 mg/mL, respectively | | |
| *X. campestris* pv. *phaseoli*, | | EO/menthol/menthone | MIC: 0.07–1.25/0.625/2.6 mg/mL, respectively | | |

**Table 1.** *Cont.*

| Target Bacterium | Species of *Mentha* | Tested Compound(s) | Activity/Toxicity | Method(s)/Dose | Reference |
|---|---|---|---|---|---|
| *Acidovorax citrulli*, | *M. × piperita* | EO/menthol/neomenthol/isopulegone/1,8-cineole | Significant inhibition at 20 µL | Disk diffusion assay | [21] |
| | | EO/menthol/neomenthol/isopulegone/1,8-cineole | Prevention of bacterial growth at 0.2% concentration | in vivo | |
| *Xanthomonas arboricola* pv. *juglandis*, | *M. × piperita M. spicata* | EOs/extracts | 5.8–3.2 mm radius of inhibition zone | Diffusion assay | [24] |
| *Xanthomonas vesicatoria*, | *M. × piperita* | EO | Strong inhibitory effect/6–12 mm zone of inhibition | Disk diffusion assay (filter paper discs of 5 mm in diameter were immersed in EO and placed onto the inoculated medium) | [22] |
| *Erwinia carotovora* subsp. *carotovora*, | | | Moderate inhibition/2–6 mm zone of inhibition | | |
| *Bacillus* sp., | | | | | |
| *Pseudomonas syringae* pv. *tomato*, | | | No effect | | |
| *P. marginalis* pv. *marginalis*, | | | No effect | | |
| *P. syringae* pv. *syringaea*, | | | No effect | | |
| *Staphylococcus aureus*, | *M. × piperita* | EO | MBC [3]: 0.57 mg/mL | Encapsulation in chitosan–nanoparticles | [23] |
| *Escerichia coli*, | | EO | MBC: 1.15 mg/mL | without encapsulation | |
| *Pseudomonas* sp., | *M. pulegium* | EO | Resistant (*Pseudomonas* genus were particularly resistant) | | [28] |
| Gram-negative bacteria, | *M. pulegium* | EO | Sensitivity | | [27] |
| *Erwinia amylovora*, | *M. arvensis* | EO | Antibacterial efficacy almost 20% 10.9 cm zone of inhibition | 1 µL/plate, diffusion assay | [30] |
| *Pseudomonas syringae* pv. *syringae*, | | Menthol | 5.0 cm zone of inhibition | | |
| *Aerococcus viridans*, *Clavibacter michiganense*, *Kocuria varians*, *Pseudomonas syringae* pathovars, *Erwinia* spp., *Xanthomonas* taxa, *Neisseria subflava*, *Agrobacterium tumefaciens*, | | | Effective inhibition on the growth of 16 from 30 strains | Disk diffusion | [34] |
| | | Menthone | Poor/it inhibited the growth of 2 from 30 strains | | |
| *C.michiganensis* subsp. *michiganensis*, *P. savastanoi* pv. *savastanoi* | *M. suaveolens* | EO | MIC: 0.78 mg/mL, drastic only against *C. michiganensis* | Disc diffusion and microdilution assay | [33] |

[1] EO, Essential Oil; [2] MIC, Minimum Inhibitory Concentration; [3] MBC, Minimum Bactericidal Concentration.

As it concerns the mode of action of EOs (and consequently of *Mentha* EOs) against bacteria, Morris [6], in a comprehensive survey, mentioned that EOs impair the biological membranes due to their lipophilic character, yet specific functional groups are also efficient. Previously, Trombetta et al. [35] said that monoterpenes act on cell membrane inducing leakage of the intracellular membrane through their action on the lipid fraction of plasma membranes, whereas Knobloch et al. [36] reported that the antimicrobial potency of EOs is owed to their solubility in the phospholipid bilayers of bacterium or fungus cells. Moreover, Cox et al. [37] mentioned that monoterpenoids affect the respiratory enzymes of fungi, which inhibits the uptake of microbial oxygen and oxidative phosphorylation. Sivropoulou [38] stated that the antibacterial action of some EOs is due to the presence of phenolic constituents. In summary, several constituents per EO may have bioactive properties and there is probably a complex mechanism with synergistic effects. In the same direction, Xu et al. [39] said that the antibacterial effects of EOs are associated with their ability to permeabilize and depolarize the cytoplasmic membrane, resulting in bacteria death. Rhouma et al. [40] explained that phenolic compounds like menthol and carvone create complexes with bacterial enzymes and proteins, thus inhibiting the bacterial proliferation.

In addition to the aforementioned activity, a literature survey by Salehi et al. [10] presented the uses of EOs and other derived extracts of the *Mentha* species as natural food preservatives against a variety of microorganisms in order to extend the shelf-life of fruits and vegetables.

### 2.2. Antifungal Activity

In recent decades, there has been increasing scientific interest in bioactive plant products, like EOs, as possible alternatives to synthetic fungicides [41–43]. This trend originates mainly from the resistance of numerous fungi to several synthetic compounds, and from the restriction of some synthetic fungicides because of their supposed entrance into the food chain [44]. Fungicides derived from the secondary metabolites of medicinal plants could be used as alternatives for pest management and are especially valuable in organic farming systems [45]. However, limited knowledge exists concerning the *Mentha* species, whereas several studies have reported on the utilization of microencapsulated *Mentha* EOs as bioagrochemicals against fungi or bacteria [23,46].

Kadoglidou et al. [47] examined both in vitro and in soil environments the effect of *M. spicata* EO and its main component carvone on growth, sporulation, and mycelium recovery of four plant pathogens: the soilborne fungi *Fusarium oxysporum* and *Verticillium dahliae*, that cause mainly wilts in cultivated plants, as well as two postharvest fungi: *Aspergillus terreus* and *Penicillium expansum*. They used a disk diffusion assay and found that the inhibitory activity (especially at the dose of 10 μL of *M. spicata* EO or of carvone per plate) showed fungistatic action against *A. terreus* and *F. oxysporum*, but fungicidal against *V. dahliae*—a pathogen very resistant to chemical agents. Moreover, Kadoglidou et al. [45] found that *M. spicata*, incorporated into the soil as dried plant material at a dose of 4% (*w/w*, plant material:soil) improved tomato tolerance against soilborne fungi. In particular, they found that plants grown in soil amended with *M. spicata* and inoculated with *Fusarium oxysporum* f.sp. *lycopersici* or *Verticillium ahlia*, did not show disease symptoms 50 days after the transplantation of inoculated tomato at a net-greenhouse, whereas the outcome of this study strongly supports that of the AUDPC values (area under diseases progress curves): for both fungi inoculation, plants grown in soil incorporation with spearmint had up to 3.5 times lower AUDPC compared to the positive controls. Moreover, *M. spicata* EO at a 10% concentration showed a complete reduction of disease incidence of *Botrytis cinerae*, one of the most significant strawberry postharvest pathogen, indicating the possible exploitation of this EO as an antifungal means of the preservation of strawberries [48].

Moreover, Domingues and Santos [25], summarizing the efficacy of *M. pulegium* EO as a biocide, found that *M. pulegium* EO is a favorable antifungal agent alternative to pesticides. Benomari et al. [49] reported the high fumigant antifungal potency of several Algerian *Mentha* oils (*M. rotundifolia*, *M. spicata*, *M. pulegium*, and *M. × piperita*) against fungi like *Botrytis cinerea*, *Monilinia laxa*, and *Monilia fructigena*, and moderate activity against *Penicillium expansum*. Their results demonstrated that the above *Mentha* EOs could be used as biological antifungal agents, providing protection on apple and pear trees from fungal infections of *Monilinia* sp. and *Botrytis cinerea*.

Guerra et al. [50] mentioned that the combinations of *M. × piperita* and *M. villora* EOs at 2.5 or 1.25 μL/mL with the simultaneous use of chitosan at 4 mg/mL strongly reduced the mycelial development and the spore germination of the following serious postharvest pathogens: *Aspergillus niger*, *Botrytis cinerea*, *Penicillium expansum*, and *Rhizopus stolonifer* in cherry tomato fruits, revealing a promising postharvest treatment for the protection from mold infections during fruit storage. These type of synergistic mixtures of *Mentha* EOs with chitosan have also been reported from de Oliveira et al. [51], who mentioned the inhibition of the *Colletotrichum* species and anthracnose development in mango fruits when the fruits are covered. Indeed, it was found that mixtures of 0.3, 0.6, or 1.25 μL/mL *M. × piperita* EO and 5 or 7.5 mg/mL chitosan strongly restrained the mycelial development and presented various additive or synergistic–inhibitory effects on the investigated *Colletotrichum* strains. Notably, disease lesion severity in mangoes coated with these blends was equal or inferior

of those observed in mangoes treated with chemical fungicides, like thiophanate-methyl and difenoconazole. Several other studies dealt with the antifungal activities of EO or extract of *M.* × *piperita* [52–54]. In particular, in a hexane extract of *M.* × *piperita*, menthol and menthone, among other constituents, are associated with antifungal activity against the seed-borne fungus in maize *Fusarium verticillioides* [53]. Another study concerning the in vitro antifungal action of *Mentha* × *piperita* EO against *Dreschlera spicifera*, *F. oxysporum* f.sp. *ciceris*, and *Macrophomina phaseolina* revealed a dose dependent action, although no fungicidal activity was observed in concentrations up to 1600 ppm [54]. However, the previous study demonstrated that 800 ppm and 1600 ppm in *D. spicifera* and 1600 ppm in *F. oxysporum* f.sp. *ciceris* caused 100% MGI.

In another study, Hanana et al. [55] assessed the antifungal activity of *M. pulegium* EO by using a disc diffusion assay, at a concentration of 0.5 mg/mL into PDA agar, against ten important plant pathogens, mainly cereals, as well as stored foods. They displayed moderate inhibitory effect on certain species of *Fusarium* genus like *F. culmorum*, *F. avenaceum*, *F. oxysporum*, *F. subglutinans*, *F. verticillioides*, *F. nygamai*, and on *Bipolaris sorokiniana*, *Botrytis cinerea*, and *Microdochium nivale*. The most impressive finding was that *M. pulegium* EO could inhibit more effectively the development of *Alternaria* sp. than the synthetic fungicides. However, Kouassi et al. [56] found that *M. pulegium* oil has poor antifungal activity against *Penicillium italicum* by measuring optical density (at 492 nm) in a micro-bioassay method with tested concentrations at 100, 500, or 1000 ppm, respectively. The above statements agree with those published by Hajlaoui et al. [57] who reported that only the high concentration (100 µL/mL) of *M. pulegium* oil caused high antifungal activity (growth inhibition 74–90.6%) against *B. cinerea*, *F. culmorum*, *F. oxysporum*, *A. niger*, *A. flavus*, and *Trichoderma* sp. The same researchers noted that the methanol extracts of the above ground parts of plant are ineffective. Silva et al. [58] demonstrated that *M. pulegium* EO highly expressed geraniol synthase gene transcripts, which is the responsible precursor enzyme for the biosynthesis of geraniol, a strong fungicidal monoterpene. This hypothesis was confirmed in vitro and in vivo against ramulosis (*Colletotrichum gossypii* South var. *cephalosporioides*), a serious fungal disease of the cotton crop, which damages leaves, stems, and bolls by decreasing fiber formation. Thus, *M. pulegium* EO inhibited the fungal growth in vitro at 1 mL/L, whereas, when it was sprayed preventively in vivo at 2 mL/L over the plants, it was reduced the early and late severity symptoms of disease by 48% and 52%, respectively. Additionally, when sprayed as curative at the same dose, it was reduced by 44% and 54% the same severity indices, respectively. Moreover, a regime of 1.5 mL/L of *M. pulegium* EO completely inhibited the fungi at 7 days. Domingues and Santos [25] stated that some compounds found in *M. pulegium* EO were photoactive, so practices employing *M. pulegium* EO as a biopesticide would potentially benefit from exposure to sunlight. This is based on a study by Matos [59], who found that the fungicidal activity of *M. pulegium* EO against *Cladosporium cucumerinum* and *Fusarium culmorum* was higher in the case of a sunlight simulator, rather than incubated in the dark.

According to Benomari et al. [49], the strongest antifungal activity of *Mentha* EOs was attributed to alcohol, aldehyde, and ketone compounds, like linalool in *M.* × *piperita*, carvone in *M. spicata*, as well as pulegone, menthone, and neo-menthol in both *M. pulegium* and *M. rotundifulia*, which showed higher antifungal potential than the oxide compounds, like piperitone oxide. Thus, regarding the *M. pulegium* EO, it is important to know the chemotype, i.e., if the main component is pulegone or piperitone oxide. However, a review by Kalemba and Synowiec [60] found that menthol was more effective than menthone. In particular, Tsao and Zho [61] found that menthol at 250 µg/mL caused 96–97% inhibition of conidial germination of *Botrytis cinerea* and *Monilia fructicola*, whereas menthone only caused 45 and 8% inhibition, respectively. Menthol was efficient at 100 µg/mL towards *M. fructicola* (mycelial growth reduction of 95%), but less active in the case of *B. cinerea* (47%). Similarly, Hussain et al. [62] found, by using disc diffusion and broth microdilution assessments, that *M. arvensis*, *M.* × *piperita*, *M. longifolia*, and *M. spicata* EOs, as well as their major components menthol, menthone, piperitenone oxide and carvone, showed

significant antimicrobial activity against the plant-pathogenic fungi. They also noted that *M. arvensis* EO exhibit higher antimicrobial activity than *M. × piperita*, due to the higher amount in menthol, which is more efficient than menthone. Specifically, they found that menthol exhibited similar inhibition (MIC by 30.8–107.7 μg/mL) to that of the standard drug fuconazole (MIC 10.4–100 μg/mL) against the abovementioned plant pathogens. Regarding *M. × piperita* oil, Beyki et al. [46] found that the encapsulation of this oil in chitosan-cinnamic acid nanogel increased the antifungal activity against *Aspergillus flavus*. Moreover, the effect of EO on the mycelial growth of *Verticillium ahlia* and *Fusarium oxysporum* was studied by Üstüner et al. [63]. The *M. longifolia* EO had 100% effectiveness on *V. ahlia* mycelium development in all doses studied. Nevertheless, *M. longifolia* oil was found to be about 30% efficient at concentration of 5 μg/cm$^2$ on mycelium growth of *F. oxysporum*, whereas at 10, 15, or 20 μg/cm$^2$ it completely hindered the development. Similarly, the growth of strains of *Rhizoctonia solani*, *Helminthosporium solani*, *Phytopthora erythroseptica*, *Fusarium coeruleum*, *Pythium ultimum*, *Phoma exigua*, and *Aspergillus flavus*, which induce potato storage diseases, were inhibited to an extended degree due to carvone, menthone, peppermint, and spearmint [64].

The antifungal activity of *Mentha* species (EOs, extracts, etc.) against phytopathogenic fungi is summarized in Table 2.

**Table 2.** Antifungal activity of *Mentha* species (EOs, extracts, etc.) against phytopathogenic fungi in the cited literature.

| Target Fungi | Species of *Mentha* | Tested Compound(s) | Activity/Toxicity | Method(s)/Doses | Reference |
|---|---|---|---|---|---|
| *Fusarium oxysporum*, | *M. spicata* | EO [1]/carvone | Moderate/Fungistatic at 10 μL/plate | In vitro, disk diffusion assay, tested doses: 1, 5, 10 μL/plate | [47] |
| *Verticillium dahliae*, | | | Very strong/Fungicidal at 10 μL/plate | | |
| *Aspergillus terreus*, | | | Strong/Fungistatic at 10 μL/plate | | |
| *Penicillium expansum*, | | | Strong/Fungistatic at 10 μL/plate | | |
| *Fusarium oxysporum* f.sp. *lycopersici*, *Verticillium dahliae*, | *M. spicata* | Dry raw material | Tomato plants recovered from the initial inoculation of both fungi 50 days after transplantation | Incorporation of dried plant material into the soil at the dose of 4% (*w/w*, plant material:soil) | [45] |
| *Botrytis cinerea*, *Monilinia laxa*, *M. fructigena*, | *M. rotundifolia M. spicata M. pulegium M. × piperita* | EOs | Strong | In vitro, disk diffusion assay | [49] |
| *Fusarium culmorum*, *F. avenaceum*, *F. oxysporum*, *F. subglutinans*, *F. verticillioides*, *F. nygamai*, *Bipolaris sorokiniana*, *Botrytis cinerea*, *Microdochium nivale*, | *M. pulegium* | EO | Moderate | In vitro, disk diffusion assay. EO dilution in 1 mL of Tween 20 (0.1% *v/v*) and then addition of 20 mL PDA | [55] |
| *Penicillium italicum*, | *M. pulegium* | EO/methanol extracts | Poor | Micro-bioassay method with tested concentration at 100, 500 or 1000 ppm | [56] |
| *Botrytis cinerea*, *F. culmorum*, *F. oxysporum*, *Aspergillus niger*, *A. flavus*, *Trichoderma* sp., | *M. pulegium* | EO | Only the high concentration (100 μL/mL) of *M. pulegium* oil caused high antifungal activity (74–90.6% MGI [2]) | In vitro | [57] |
| | | Methanol extracts | Methanol extracts were not effective | In vitro | |
| *Colletotrichum gossypii* South var. *cephalosporioides*, | *M. pulegium* | EO | Strong | 1.0 and 1.5 mL/L of EO completely inhibited the fungi in vitro and in vivo, respectively | [58] |
| *Cladosporium cucumerinum*, *Fusarium culmorum*, | *M. pulegium* | EO | Higher fungicidal activity of EO was when exposed to a sun light simulator, rather than incubated in the dark | | [59] |

**Table 2.** *Cont.*

| Target Fungi | Species of *Mentha* | Tested Compound(s) | Activity/Toxicity | Method(s)/Doses | Reference |
|---|---|---|---|---|---|
| *Botrytis cinerea,* | | Menthol and menthone | 96% and 45% inhibition of conidial germination, respectively | 250 µg/mL | [61] |
| | | Menthol | 47% MGI | 100 µg/mL | |
| *Monilia fructicola,* | | Menthol and menthone | 97% and 8% inhibition of conidial germination, respectively | 250 µg/mL | |
| | | Menthol | 95% MGI | 100 µg/mL | |
| *Alternaria alternata, Alternaria solani, Aspergillus flavus, Aspergillus niger, Fusarium solani, Rhizopus solani, Rhizopus* spp., | *M. arvensis M. × piperita M. longifolia M. spicata* | EOs/their major components menthol, menthone, piperitenone oxide and carvone, respectively | Notable antifungal activity | Disk diffusion/broth microdilution | [62] |
| *Aspergillus flavus,* | *M. × piperita* | EO | Enhancement of antifungal activity | Encapsulation in chitosan–cinnamic acid nanogel | [46] |
| *Verticillium dahliae,* | *M. longifolia* | EO | 100% MGI at all concentrations | In vitro/5, 10, 15 and 20 µg/cm² | [63] |
| *Fusarium oxysporum,* | | | 100% MGI at 10–20 µg/cm² | | |
| *Rhizoctonia solani, Helminthosporium solani, Phytopthora erythroseptica, Fusarium coeruleum, Pythium ultimum, Phoma exigua, Aspergillus flavus,* | *M. × piperita M. spicata* | Carvone/ Menthone/ EOs | 100% MGI in the majority of strains | In vitro, 100 µL of pure oils and 0.1, 1, 10, 100, and 1000 ppm of constituents into each petri plate) | [64] |
| *Aspergillus niger, Botrytis cinerea, Penicillium expansum, Rhizopus stolonifera,* | *M. × piperita M. × villora* | EOs with chitosan | Strong inhibition of MGI and spore germination | 4 mg/mL chitosan + 1.25 or 2.5 µg/mL EOs | [50] |
| *Colletotrichum* strains, | *M. × piperita* | EO with chitosan | 100% MGI except of the mixture 5 mg/mL chitosan + 0.3 mL/mL EO | 5 and 7.5 mg/ML chitosan + 0.3, 0.6 or 1.25 µL/mL EO | [51] |
| *Fusarium verticillioides MRC 826,* | | Limonene/ Menthol/ Menthone/ Thymol | | Semisolid agar antifungal susceptibility technique. Concentrations: 25, 50, 75, 150, 200, 250, 500 and 1000 µL/L | [53] |
| *Dreschlera spicifera, Fusarium oxysporum* f.sp. *ciceris, Macrophomina phaseolina* | *M. × piperita* | EO | Dose dependent activity 100% MGI at 800 and 1600 ppm in some fungi | Petri plates assays in potato dextrose agar. Concentrations: 100, 200, 400, 800, 1600 ppm | [54] |

[1] EO, Eseential Oil; [2] MGI, Mycelium Growth Inhibition.

The biological action of EOs is attributed to the structural and functional group of their main components. Concerning the mode of action against fungi, Gholamipourfard et al. [52] said that cyclic monoterpene menthol—one of the main constituents of *Mentha* EOs—contributes significantly to their biological activity. Additionally, Ait-Ouazzou et al. [65] found that monoterpenes are key factors in the structural disorganization of cell membranes, leading to depolarization and chemical or physical changes, which disturb fungal metabolic activities.

### 2.3. Yeast Diseases Management

There are several works in the literature that report on the effectiveness of *Mentha* EOs or their extracts against yeasts like species of the genus *Candida* or *Saccharomyces cerevisiae*. Nevertheless, these pathogens mainly colonize humans, and do not concern the agricultural sector. Consequently, they are not analyzed in the current chapter. For instance, the following investigations related to the effectiveness of *Mentha* (*M. suaveolens, M. longifuolia, M. × piperita*) EOs or extracts against yeasts *C. albicans* 3248, *C. albicans* 3993, *C. kruseii, C. glabrata, S. cerevisiae,* are cited in: Fancello et al. [66]; Abdelli et al. [67]; Ghazghazi et al. [68]; Sarac and Ugur [28]; Mahboubi and Haghi [69]; Riahi et al. [70]; Al-Bayati [71]; Oumzil et al. [72].

However, in an earlier study by Conner and Beuchat [73] on the antimycotic properties of peppermint oil, they demonstrated activity against several food spoilage yeasts. In partic-

ular, they referred to inhibition zones of 5–9 cm against *Geotrichum candidum*, *Metchnikowia pulcherima*, *Rhodotorula rubra*, and *Torulopsis glabrata*. Peppermint oil caused a delay in the appearance of their pseudomycelium from the normal appearance almost at 18 days.

Concerning the mode of action against yeasts, Ferreira et al. [74] demonstrated that *M. × piperita* EO induces apoptosis in yeast, whereas lethal cytotoxicity is due to the elevated amount of intracellular reactive oxygen species, mitochondrial fragmentation, and chromatin condensation, while remaining intact the plasma membrane.

### 3. Target: Animals

Several *Mentha* EOs have interesting activity, particularly against insects, acarines, and nematodes.

#### 3.1. Instecticidal Activity

According to Isman [75], with regard to agricultural pest management, although plant based insecticides are well adapted in conditions of developed countries for organic food production, they could be equally in the production and postharvest protection of food products in developing countries. Hence, the following literature data reported on the *Mentha* species.

Studies focusing on the effectiveness of the *Mentha* species against insect/pests with agricultural interest are relatively large in number. The potential of different Mentha species on insect control has been assessed by running adulticidal, larvicidal, and growth/reproduction inhibition bioassays. Repellent properties of various *Mentha* EOs and extracts have been verified, whereas relevant research is mainly focused on pests belonging to coleoptera and diptera species [76].

Kumar et al. [76] studied the fumigant and repellent activity of *Mentha* EOs towards several stored grain pests such as *Tribolium castaneum*, *Sitophilus oryzae*, *Acanthoscelides obtectus*, etc., and vectors (e.g., mosquitoes). However, only a few studies have been carried out regarding the larvicidal and growth/reproduction regulatory activities of *Mentha*. Additionally, there is a lack of investigation concerning product development and the assessment of its effectiveness in real field conditions.

Domingues and Santos [25], in a comprehensive survey, reviewed the insecticidal properties of *M. pulegium*, concluding that its EO may be utilized insects' control instead of an insecticidal program. Generally, Lamiaceae EOs may restrain aphids dwelling on these plants influencing the aphids guastatory and/or olfactory sensation, whereas carvone in spearmint may be the main factor causing antifeeding and settling inhibitory activity against the green peach aphid (*Myzus persicae*) [77]. Moreover, *M. pulegium* was proven efficient in inhibiting *Sitophilus zeamais* reproduction, an insect which is the main reason for stored grains destruction, among them maize [78]. They found that *M. pulegium* oil at a minimum concentration of 0.16 μL/cm$^2$ provided adult mortality at 24 h, while no progeny production was achieved. These results are due to the capacity of *M. pulegium* to obstruct oviposition behavior or because of its toxicity to larvae [78]. The same findings are supported by Rocha et al. [79], who investigated the capacity of *M. pulegium* for mosquito control. Low doses of 2.5–5 μL EO/mL acetone of *M. pulegium* inhibited the wheat weevil *Sitophilus granaries*. Inhalation was the most efficient technique, followed by the ingestion, and finally by the contact technique, which was successful at a higher dose of 20 μL EO/mL acetone [67]. Lougraimzi et al. [80] studied the insecticidal effect of EO and powdered *M. pulegium* leaves against *Sitophilus oryzae* and *Tribolium castaneum*. They concluded that 0.16 μL/cm$^2$ of oil administered within 24 h caused 100.0% mortality for both insects by contact, whereas 20 μL/L air resulted in 100% fumigant mortality for *S. oryzae* and *T. castaneum* at 24 and 48 h, respectively, by inhalation. Finally, 0.25 μL/g at 48 h for both insects caused 100% mortality by ingestion. Sohani [81] found that EOs vapor of both *M. pulegium* and *M. viridis* leaded the maximum mortality in 2 μL/L air dose after 24 h of exposure in cotton whitefly (*Bemisia tabaci*). Moreover, *M. pulegium* oil resulted in 100% mortality of *Mayetiola destructor*, the most significant wheat pest in Morocco [82].

*Bactrocera* (*Dacus*) *oleae* is the most serious agricultural pest for olive trees, and causes severe annual damages in olive crops. Pavlidou et al. [83] assessed the susceptibility of larvae of *B. oleae* and *Drosophila melanogaster* in *M. pulegium* oil, concluding that the $LD_{50}$ were 0.22 and 2.09 µL/L, respectively. In the same study, for *B. oleae* the $LD_{50}$ were 0.9 and 0.13 µL/L for pulegone and menthone, respectively, whereas for *D. melanogaster*, they were 0.17 and 1.29, respectively. Furthermore, fumigant toxicity assays on the second and third larval instars of the specific defoliator pest of lettuce *Anarta trifolii* (Hufnagel) exhibited the highest sensitivity to *M. pulegium* oil, with $LC_{50}$ at 0.41 and 0.80 µL/L air, and $LC_{90}$ of 0.88 and 9.14 µL/L air, respectively [84]. *M. pulegium* concentrations of 0.89, 1.34, and 2 µL/L showed the maximum antifeedant activity on the fourth instar larvae, 47.88%, 31.80%, and 11.89% eaten leaf surface, respectively. Salem et al. [85] referred to the potential fumigant of *M. pulegium* oil impact against *Lasioderma serricorne* with $LC_{50}$ of 8.46 µL/L air, and remarkable pest repellent efficacy 60% after 24 h of exposure against *Tribolium castaneum* at rate of 0.078 µL/cm$^2$. *M. pulegium* and *M. piperata* oils, in combination with other biological pest control methods, i.e., the utilization of *Lecanicillium muscarium* fungus, exhibited an additive result against the aphid *Aphis gossypii*, a polyphagous aphid on watermelon, cotton, and vegetables [86]. In this case, $LC_{50}$ values of *M. piperata* and *M. pulegium* oils were determined at 15.25 and 23.13 µL/L air, respectively.

Kimbaris et al. [87] studied the insecticidal activity of *M. spicata* and *M. pulegium* EOs, and their major components: iso-menthone, pulegone, carvone, piperitone, piperitone oxide, and piperitenone oxide. They found that their EOs were effective antifeedants against *Leptinotarsa decemlineata* and *Myzus persicae*, followed by *Spodoptera littoralis*. Concerning their major compounds, *L. decemlineata* was the most sensitive and was strongly affected by piperitenone and piperitone epoxide, whereas *S. littoralis* was affected by piperitone epoxide and pulegone. A similar approach was obtained by Santana-Méridas et al. [88], who reported antifeedant effects of *M. pulegium* (especially) and *M. spicata* oils from Morocco against *S. littoralis*, *M. persicae* and *Rhopalosiphum padi*. Moreover, pulegone epoxide, carvone, carvone epoxide, piperitenone oxide, and piperitone demonstrated considerable antifeedant, toxic, and repellent activity against the chewing and sucking insect like *Alphitobius diaperinus*, a widespread pest affecting massively chicken and broiler houses [89].

In a contact bioassay, *M. longifolia* subsp. *capensis* EO at the dose of 0.50 µL/g on maize seeds resulted in 100% mortality of *Sitophilus zeamais*, in comparison to lower than 10% mortality using dose of 0.125 µL/g [90]. In the fumigation bioassay of the same study, the EO of *M. longifolia* demonstrated moderate fumigation toxicity against the same coleopter (% cumulative mortality > 60%, at rates of 24 and 32 µL of oil/L air). Moreover, Abbas and Javad [91] reported that adults of *Tribolium castaneum* were eradicated by *M. longifolia* oil at 13.05 mL/L air $LC_{50}$ value by fumigant bioassay.

Kumar et al. [92] found that *M. arvensis* oil exhibited potent insecticidal activity against the insect of stored chickpea *Callosobruchus chinensis*. More precisely, the oviposition by *C. chinensis* was totally controlled at 10 µL/L, while F1 emergence was absolutely hindered at 200 µL/L. Notably, in situ experiments showed 94.05% efficacy of *M. arvensis* oil over 90.75% of the organophosphate insecticide malathion. The insecticidal action of *M. arvensis* has been also investigated. Specifically, Lee et al. [93] mentioned that, among sixteen spices and medicinal plants, *M. arvensis* var *piperascens* oil presented the most potent toxicity ($LC_{50}$ = 45.5 µL/L of air) against rice weevil *Sitophilus oryzaeha*. Among its major components, menthone was the most active against *S. oryzaeha* ($LC_{50}$ = 12.7 µL/L of air), followed by linalool ($LC_{50}$ = 39.2 µL/L of air), and alpha-pinene ($LC_{50}$ = 54.9 µL/L of air). Varma and Dubey [94] studied the *M. arvensis* oil as fumigant against *Sitophilus oryzae* and *Tribolium castaneum* (two serious storage pests that cause damage to food products), and reported that fumigation of wheat grains with 600 ppm of *M. arvensis* oil completely inhibited the insects. The antifeedant activity of *M. arvensis* oil towards the feeding deterrence index (FDI) was also evaluated at 94% against *Callosobruchus chinensis* [92], whereas it was at 15–42% against the onion thrip *Thrips tabaci* [95]. It was reported that *M. arvensis* oil decreased at 67.5%

over control the acetylcholinesterase activity of *T. castaneum* after 24 h of fumigation in laboratory assay [96].

The inhibitory activities of *M. × piperita* and their main constituents menthone and menthol against drosophila (*Drosophila suzukii*) were examined by Park et al. [97]. They mentioned that the $LD_{50}$ (mg/L) values of *M. × piperita* oil, menthone, and menthol was 3.87, 5.76, and 1.88 against males and 4.10, 5.13, and 1.94 against female insects, respectively. Toxicity of *M. × piperita* oil and menthone against the red flour beetle *T. castaneum* was evaluated after fumigation for 24 h in 250 mL conical flasks [98]. The $LD_{50}$ values were estimated at 25.8 μL/L and 8.5 μL/L air for *M. × piperita* oil and menthone, respectively. Çam et al. [99] investigated the fumigant effect of *M. × piperita*, *M. spicata*, and *M. villoso-nervata* towards the granary weevil (*Sitophilus granaries*). *M. villoso-nervata* oil was the most toxic among the oils, exhibiting 90% mortality of adults by fumigant bioassay. Additionally, the EO main compound carvone showed 100% mortality at 24 h of exposure with a 0.024 μL/mL $LC_{50}$ value. The above results demonstrated that *M. villoso-nervata* and carvone were potentially effective in granary weevil control. Koundal et al. [100] evaluated the insecticidal activitiy of *M. × piperita*, *M. spicata*, and *M. longifolia* towards the larvae of diamondback moth *Plutella xylostella*, an insect pest of cruciferous crops. It was revealed that *M. longifolia* was the most toxic ($LC_{50}$ = 1.06 mg/mL) to sthe econd instar larvae of *P. xylostella* applying the residual toxicity bioassay, followed by *M. × piperita* ($LC_{50}$ = 1.37 mg/mL). Moreover, *M. × piperita* and *M. spicata* exhibited potential repellent ($RC_{50}$ = 1.33 mg/mL) and feeding deterrence activity (66.07%) to the third instar larvae, respectively. Another study by Saeidi and Mirfakhraie [101] stated that $LC_{50}$ of *M. × piperita* oil was 25.70 μL/L air against *Callosobruchus maculates*, a store pest of leguminous seeds, while the persistence test revealed that EO of *M. × piperita* on *C. maculatus* adults was 5.44 days.

Souza et al. [102] stated that *M. spicata* oil presented fumigating properties to promote the control of *Rhyzopertha dominica*, affecting the stored maize, by showing $LC_{50}$ value of 27.52 mL/L of air. Moreover, *M. spicata* oil induced 100% mortality in the insect pest *Callosobruchus chinensis*, with an $LC_{50}$ value of 0.003 μL/mL of air 24 h after fumigation treatment, and 100% repellence at 0.025 μL/mL air concentration [103]. *M. spicata* oil, at a dose 0.1 μL/mL of air, was a potent fumigant, recording 98.46% oviposition deterrency, 100% ovicidal, 88.84% larvicidal, 72.91% pupaecidal, and 100% antifeedant activity against *C. chinensis*. Eliopoulos et al. [104] found that *M. spicata* was highly effective towards two serious stored products pests, *Ephestia kuehniella* and *Plodia interpunctella*, with significant mortality over 80% after exposure to low doses such as 2.5 mL/L. Notably, egg mortality was 56–60%, larval mortality never exceeded 18%, whereas pupae displayed mortality as high as 28%. *M. spicata* oil by fumigation exhibited 259.73 and 75.31 ppm $LC_{50}$ value towards adults and fourth instar larvae of potato beetle *Leptinotarsa decemlineata*, respectively, whereas 39.26% feeding deterrent index against the adults at 16 ppm [105]. According to Aslan et al. [106], *M. spicata* subsp. *tomentosa* and *M. spicata* var. *formasa* oils caused 100% mortality of *Sitophilus granarius* at 1 mL/L air and exposure for 36 and 48 h, respectively, and strong mortality of adults at 0.5 mL/L air and an exposure period of 48 h. Recently, a significant fumigant toxicity of *M. spicata* EO and its dominant constituents carvone, dihydrocarvone, and limonene against termites (*Reticulitermes dabieshanensis*), due to the inhibition of AChE activity, was reported [107].

Allahvaisi [108] referred that 1.75 mL of *M. viridis* oil per 0.5 mL acetone dose had the most repellent effectiveness on *Sitophilus granarius* (63.81%).

Moreover, Yakhlef et al. [109] found that the $LC_{50}$ of *M. rotundifolia*'s EO against *Sitophilus granarius* and *Tribolium confusum* in fumigant and repellent bioassays was 1.072 μL/mL and 1.530 μL/mL, respectively.

The insecticidal activity of *Mentha* species (EOs, extracts, etc.) against insects with agricultural interest is summarized in the Table 3.

**Table 3.** Insecticidal activity of *Mentha* species (EOs, extracts, etc.) against insects with agricultural interest in the cited literature.

| Target Insect | Species of *Mentha* | Tested Compound(s) | Activity/Toxicity | Method(s)/Dose | Reference |
|---|---|---|---|---|---|
| Green peach aphid (*Myzus persicae*), | Spearmint | Carvone | Antifeeding and settling inhibitory | Aphids diets with or without EO into plastic vessels | [77] |
| *Sitophilus zeamais*, | *M. pulegium* | EO [1] | Effect on reproduction adult mortality within 24 h | 0.16 μL/cm$^2$ | [78] |
| *Sitophilus granaries*, | *M. pulegium* | EO | Inhalation and Ingestion: 100% mortalityBy contact: LD$_{50}$ [2] = 9.11 ± 2.53 μL/mL | Inhalation: 2.5–5 μL EO/mL acetone | [67] |
| *Sitophilus oryza*, *Tribolium castaneum*, | *M. pulegium* | EO and powder | By contact: 100.0% mortality | 0.16 μL/cm$^2$ | [80] |
| | | | Fumigant: 100% mortality | 20 μL/L air | |
| | | | Ingestion: 100% mortality | 0.25 μL/g | |
| *Bemisia tabaci*, | *M. pulegium* *M. viridis* | EO | High mortality | 2 μL/L air | [81] |
| *Mayetiola destructor*, | *M. pulegium* | EO | 100% adult mortality | 20 μL/L air | [82] |
| *Bactrocera* (Dacus) *oleae*, | *M. pulegium* | EO | LD$_{50}$: 0.22 μL/L | 1 mL diluted in acetone 2% *v/v* and applied on filter paper in petri dishes | [83] |
| | | Pulegone/ | LD$_{50}$: 0.9 μL/L | | |
| | | Menthone | LD$_{50}$: 0.13 μL/L | | |
| *Drosophila melanogaster*, | | EO | LD$_{50}$: 2.09 μL/L | | |
| | | Pulegone/ | LD$_{50}$: 0.17 μL/L | | |
| | | Menthone | LD$_{50}$: 1.29 μL/L | | |
| *Anarta trifolii*, | *M. pulegium* | EO | 2nd larval instar LC$_{50}$ [3]: 0.41 μL/L air 3rd larval instar LC$_{50}$: 0.80 μL/L air 2nd larval instar LC$_{90}$ [4]: 0.88 μL/L air 3rd larval instar LC$_{90}$: 9.14 μL/L air | 0.89, 1.34, 2 μL/L | [84] |
| *Lasioderma serricorne*, *Tribolium castaneum*, | *M. pulegium* | EO | LC$_{50}$ of 8.46 μL/L air 60% repellent activity | 0.078 μL/cm$^2$ | [85] |
| *Aphis gossypii*, | *M. pulegium* *M. piperata* | EO | LD$_{50}$: 23.13 μL/L LD$_{50}$: 15.25 μL/L | | [86] |
| *Leptinotarsa decemlineata*, *Myzus persicae*, *Spodoptera littoralis*, | *M. spicata* *M. pulegium* | EOs (and iso-menthone, pulegone, carvone, piperitone, piperitone oxide, piperitenone oxide) | Feeding inhibition: 75.3–84.6% Feeding inhibition: 83% Feeding inhibition: 87.6–89.9% Feeding inhibition: 74% Feeding inhibition: 75.1–80.8% Feeding inhibition: 51.2% | | [87] |
| *Myzus persicae*, *Spodoptera littoralis*, *Rhopalosiphum padi*, | *M. pulegium* *M. spicata* *M. pulegium* *M. spicata* *M. pulegium* *M. spicata* | EOs | % SI [5]: 77.9 μg/cm$^2$ % SI: 48.9 μg/cm$^2$ % FI [6]: 100 μg/cm$^2$ % FI: 48.9 μg/cm$^2$ % SI: 85.3 μg/cm$^2$ % SI: 43.6 μg/cm$^2$ | | [88] |
| *Alphitobius diaperinus*, | | Synthetic pulegone epoxide/ Carvone/ Carvone epoxide/ Piperitenone oxide/ Piperitone | Repellent and strong antifeedants | | [89] |
| *Sitophilus zeamais*, | *M. longifolia* subsp. *capensis* | EO | 100% mortality | 0.50 μL/g | [90] |
| *Tribolium castaneum*, | *M. longifolia* | EO | Strong activity LC$_{50}$: 13.05 μL/L air | Fumigation | [88] |
| *Callosobruchus chinensis*, | *M. arvensis* | EO | 10 μL/L completely controlled the oviposition In situ: 94.05% protection of the chickpea from insect | | [92] |
| *Sitophilus oryzaeha*, | *M. arvensis* var. *piperascens* | EO/ Menthone/ Linalool/ Alpha-pinene | LC$_{50}$: 45.5 μL/L of air LC$_{50}$:12.7 μL/L of air LC$_{50}$: 39.2 μL/L of air LC$_{50}$: 54.9 μL/L of air | | [93] |
| *Sitophilus oryzae*, *Tribolium castaneum*, | *M. arvensis* | EO | 100% inhibition | Fumigation with 600 ppm | [94] |
| *Thrips tabaci*, | *M. arvensis* | EO | Feeding deterrence Index: 15–42% | | [95] |
| *Tribolium castaneum*, | *M. arvensis* | EO | Inhibition of acetylcholinesterase activity about 67.5% | Fumigation with sub-lethal concentration | [96] |

**Table 3.** *Cont.*

| Target Insect | Species of *Mentha* | Tested Compound(s) | Activity/Toxicity | Method(s)/Dose | Reference |
|---|---|---|---|---|---|
| *Drosophila suzukii,* | *M. × piperita* | EO/ Menthone/ Menthol | $LD_{50}$: 3.87 mg/L against males $LD_{50}$: 4.1 mg/L against females $LD_{50}$: 5.76 mg/L against males $LD_{50}$: 5.13 mg/L against females $LD_{50}$: 1.88 mg/L against males $LD_{50}$: 1.94 mg/L against females | | [97] |
| *T. castaneum,* | *M. × piperita* | EO/ Menthone | 25.8 µL/L air 8.5 µL/L air | Fumigation for 24 h | [98] |
| *Sitophilus granaries,* | *M. × piperita* *M. spicata* *M. villoso-nervata* | EOs | The most toxic (90% mortality) was *M. villoso-nervata* oil | Fumigation for 24 h with 0.024 µL/mL $LC_{50}$ value | [99] |
| *Plutella xylostella,* | *M. × piperita* *M. spicata* *M. longifolia* | EOs | $LC_{50}$ = 1.37 mg/mL $RC_{50}$ [7] = 1.33 mg/mL $LC_{50}$ = 1.06 mg/mL | Residual toxicity bioassay | [100] |
| *Callosobruchus maculates,* | *M. × piperita* | EO | $LC_{50}$: 25.70 µL/L | | [101] |
| *Rhyzopertha dominica,* | *M. spicata* | EO | $LC_{50}$: 27.52 mL/L of air | | [102] |
| *Callosobruchus chinensis,* | *M. spicata* | EO | $LC_{50}$: 0.003 µL/mL of air 100% repellency 98.46% oviposition deterrency 100% ovicidal activity 88.84% larvicidal activity 72.91% pupaecidal activity 100% antifeedant activity | Fumigation 0.025 µL/mL air 0.1 µL/mL air | [103] |
| *Ephestia kuehniella,* *Plodia interpunctella,* | *M. spicata* | EO | adult mortality: 80% egg mortality: 56–60% larval mortality < 18% pupae mortality < 28% | 2.5 mL/L | [104] |
| *Leptinotarsa decemlineata,* | *M. spicata* | EO | $LC_{50}$: 259.73 ppm for adults $LC_{50}$: 75.31ppm for 4th instars larvae | Fumigation | [105] |
| *Sitophilus granaries,* | *M. spicata* subsp. *tomentosa* *M. spicata* var. *formasa* | EOs | 100% mortality | 1 mL/L air | [106] |
| *Sitophilus granaries,* | *M. viridis* | EO | 63.81% repellent effectiveness | 0.5 mL acetone dose | [108] |
| *Sitophilus granaries,* *Tribolium confususm,* | *M. rotundifolia* | EO | $LC_{50}$: 1.072 µL/mL air $LC_{50}$: 1.530 µL/mL air | Fumigant toxicity in glass jar and repellency bioassay with filter paper disk in petri | [109] |
| *Reticulitermes dabieshanensis* | *M. spicata* | EO/ Carvone/ Dihydrocarvone/ Limonene | $LC_{50}$: 0.134–0.213 µL/L $LC_{50}$: 0.045–0.115 µL/L $LC_{50}$: 0.096–0.213 µL/L $LC_{50}$: 2.468–5.149 µL/L, Strong acetylocholinesterase inhibition | Fumigant toxicity in 1 L glass jar with 0.03–6 µL of tested compounds determined at 15, 20, 25 and 30 °C | [107] |

[1] EO, Essential Oil; [2] $LD_{50}$, Lethal Dose killed 50% of population; [3] $LC_{50}$, Lethal Concentration killed 50% of population; [4] $LC_{90}$, Lethal Concentration killed 90% of population; [5] % SI, Percent Setting Inhibition (100 µ/cm$^2$ for EOs); [6] % FI, Percent Feeding Inhibition (100 µ/cm$^2$ for EOs).[7] $RC_{50}$, Repellent Concentration repelled 50% of population.

With regard to the mode of insecticidal activity of EOs, researchers consider that the botanical insecticides based on EOs show variable target activities on insect pests. Kumar et al. [76] suggested that the observed repellent, antifeedant, and growth regulation effectiveness is probably due to the action of EOs and their compounds on biochemical processes, which explicitly disrupt the endocrinologic balance of insects. The lipophilic properties of EOs promote their intervention with basic metabolic, biochemical, physiological, and behavioral functions of insects [110]. Some researchers proposed as a potential mode of action of *Mentha* EOs the inhibition of acetylcholinesterase activity (AChE), associated with an oxidative imbalance [96,111]. Similarly, restricting the AChE synthesis by EOs affects cholinergic synapses in insects and higher animals [112]. Nevertheless, Lee et al. [93] state that insect toxicity and AChE inhibition are not correlated. Some investigators also reported that exposing the insects to the EOs caused a breakdown of their nervous systems [113]. The octopaminergic system—which is crucial as a neurotransmitter—neurohormone, and neuromodulatorin invertebrate systems, is the main target site of EOs [114]. Previously, Pare and Tumlinson [115] stated that the mortality effect of EOs on insects is due to the penetration of their volatile constituents through the insects' respiratory system, thus causing abnormal breathing, which leads to asphyxiation and final death.

### 3.2. Acaricidal Activity

*Mentha* EOs possess acaricidal activity against many plant feeding mites like *Tetranychus urticae*, *T. turkestani*, and *T. cinnabarinus*, and towards the stored grains mites like *Tyrophagus putrescentiae* as well.

Domingues and Santos [25] stated that *M. pulegium* oil increases the economic impact that they may have on agriculture, since it can also be used as an acaricidal. Generally, the maturity of the acarine affect the acaricidal activity of EOs; thus, the young stages (eggs and larvae) are more susceptible than mature ones [116]. Contrary to this, Attia et al. [117] stated that the maturity of the acarines is not associated with the mortality percentage and the acaricidal activity of EOs. Thus, Pavela et al. [118] stated that *M. pulegium* oil affects some of the most polyphagous pests, such as *Tetranychus urticae*, which damages many vegetables and ornamental plants. Similarly, Topuz et al. [119] tested in vivo *M. pulegium* oil for its fumigant toxic, and development-reproduction-inhibiting activities against *T. urticae*. The results of the study revealed that, *M. pulegium* oil was the most drastic oil against all the tested biological stages ($LC_{50}$ = 0.60 μL/L air for eggs, 0.60 μL/L air for larvae and 0. 49 μL/L air for adult females). Previously, Topuz et al. [116] found that *M. pulegium* oil at concentration of 4 μL/L air at 14 days caused an 89.3% and 72.9% decrease in the *T. cinnabarinus* larva/nymph and adult populations, respectively. Zandi-Sohani and Ramezani [120] found that quantity of 20 μL/L of *M. pulegium* or *M. viridis* oils caused 100% mortality of the strawberry spider mite *T. turkestani* after 24 h, whereas the $LC_{50}$ of females was 14.5 and 15.3 μL/L, for the two oils, respectively.

Jeon and Lee [121] found that the $LD_{50}$ value of *M. arvensis* oil in laboratory bioassays was 3.41 μg/cm$^2$ and this was about 3.52-fold more active than the synthetic acaricide benzyl benzoate against the stored food acarine *Tyrophagus putrescentiae*.

Similarly, Park et al. [122] stated that *M. × piperita* oil showed a positive acaricidal effect against *T. putrescentiae* compared to synthetic acaricide benzyl benzoate. The $LD_{50}$ of *M. × piperita* oil was 2.72 and 1.87 μg/cm$^2$ for the fumigant and petri dish bioassays, respectively, whereas the relative toxicity was around four times greater compared to benzyl benzoate.

Isman [75] found that menthol is widely used for the fumigation of beehives to manage the Varroa mite (*Varroa jcobsoni*) and the tracheal mite (*Acarapis woodi*), two honeybee parasites with economic importance. Previously, Delaplane [123] (1992) stated that menthol derived from peppermint is extensively used for these parasites in North America, while Floris et al. [124] declared that thymol is mostly used for the same purpose in Europe.

### 3.3. Nematicidal Activity

The following studies showed encouraging results regarding the utilization of *Mentha* EOs or some of their major constituents as a nematicidal.

Kimbaris et al. [87] demonstrated a strong nematicidal activity of *M. spicata* and *M. pulegium* compounds iso-menthone, pulegone, carvone, piperitone, piperitone oxide, and piperitenone oxide against root-knot nematode (*Meloydogine javanica*). The strongest nematicidal agent against *M. javanica* was achieved by piperitenone epoxide with similar $LC_{50}$ and $LC_{90}$ values (0.04 and 0.05 mg/mL, respectively), followed by piperitone epoxide, piperitenone, and carvone. Previously, Caboni et al. [125] reported on the nematicidal activity of aqueous extracts and EOs of *M. × piperita*, *M. spicata*, and *M. pulegium* against *M. incognita*. The aqueous extracts were more potent, and the $EC_{50/72h}$ values were estimated at 1005, 745, and 300 mg/L for *M. × piperita*, M. pulegium, and *M. spicata*, respectively. *M. spicata* EO was the sole that exhibited a nematicidal activity ($EC_{50/72h}$ at 358 mg/L). Menthofuran and carvone presented $EC_{50/48h}$ values 127 and 730 mg/L, respectively. Moreover, salicylic acid, which was present in the water extracts, exhibited $EC_{50}$ values at 24 and 48 h of 298 and 288 mg/L, respectively. Furthermore, a rich in carvone *M. spicata* EO showed a significant nematicidal effect on *M. javanica*, although a *M. pulegium* oil from Morocco rich in pulegone was ineffective against the same nematode [88]. Moreover, *M. pulegium* oil was not significantly effective on *Bursaphelenchus xylophilus* [126], although

considerable nematicidal action was reported against *B. xylophilus* [127]. Kimbaris et al. [87] suggested that probably minor compounds can affect the EOs action, since the main compounds of *M. pulegium* oil (piperitone and pulegone), when tested separately, have shown nematicidal effects against *M. javanica*. Carvone exhibited in vitro nematicidal action against *M. javanica*. Specifically, Oka et al. [128] reported that *M. rotundifolia* and *M. spicata* were among the twelve EOs that blocked more than 80% of juveniles of the root-knot nematode *M. javanica* at a concentration of 1000 mul/L, and they also inhibited nematode hatching. These oils incorporated in sandy soil at concentrations of 100 and 200 mg/kg decreased the root galling of cucumber seedlings in pot experiments. The major EO component, carvone, immobilized the juveniles and inhibited hatching at >125 mul/L in vitro. Additionally, carvone in a mixture with sandy soil at concentrations of 75 and 150 mg/kg eliminated root galling of cucumber seedlings, whereas the nematicidal action of the EOs and their compounds was verified at 200 and 150 mg/kg, respectively, in pots experiments.

Pandey and Kalra [129] found a considerable inhibition in hatching eggs of *M. incognita* occurred in the aqueous extracts of vermicompost produced from wastes of *M. arvensis*, followed by *M. viridis*. In the same study, in a pot experiment, vermicomposts of *M. arvensis* effectively reduced the root-knot infection in tomato. It is notable that when Khanzada et al. [130] evaluated thirteen mints for the occurrence of nematode fauna associated with their rhizospheres, they found that no plant parasitic nematode was found associated with field mint (*M. arvensis*), which can further be investigated for its role as nematode repellent, and can be used either as mulch or intercropping.

The nematicidal activity of *M. canadensis* oil and its major constituents towards second-stage juveniles of the seed-gall nematode (*Anguina tritici*), citrus nematode (*Tylenchulus semipenetrans*), root-knot nematode (*M. javanica*), and pigeon-pea cyst-nematode (*Heterodera cajani*), was evaluated by Sangwan et al. [131]. They found that the $LC_{50}$ of both *M. canadensis* oil and menthol was considerably higher than either eugenol-rich or eugenol-free clove oil. Menthol was moderately active against *T. semipenetrans* and *M. javanica*.

The mode of action of EOs and their constituents against nematodes has not yet been clarified. However, illumination concerning the mode of action of EOs and their components could provide valuable data about the most suitable formulation and delivery systems [132].

## 4. Target: Plant (Weeds and Crops)

### 4.1. Herbicidal Activity

Kadoglidou [133] stated that *M. spicata* oil (at doses of 1, 2.5, 5, and 10 µL/petri) and its main component carvone caused strong in vitro inhibition of germination, growth, and of fresh biomass of the weeds *Phalaris paradoxa* and *Datura stramonium*, and minor but significant inhibition at the same parameters of the weeds *Abutilon theophrasti* and *Oryza sativa*. The most pronounced inhibition on the growth of *P. paradoxa* and *D. stramonium* have $I_{50}$ values of approximately 2.8 µL/petri for EO of *M. spicata* and 2.2 µL/petri for carvone. Similarly, Azirak and Caraman [134] studied the effect of *M. spicata* oil (concentration of 3, 6, 10, and 20 µL/petri) on the in vitro germination of some common weed species (*Alcea pallida*, *Amaranthus retroflexus*, *Centaurea salsotitialis*, *Raphanus raphanistrum*, *Rumex nepalensis*, *Sinapis arvensis*, and *Sonchus oleraceus*). They concluded that *M. spicata* oil demonstrated great inhibitory effect against weed seeds, even at reduced concentration. The major compound carvone was investigated for seed germination at four concentrations (62.5, 125, 250, and 500 µg/mL) against the same weeds, revealing high inhibition, even at the low concentrations. Similarly, *M. spicata* oil showed allelopathic action and inhibited the germination of *Amaranthus retroflexus*, *Echinochloa crus-galli*, *Oryza sativa*, *Portulaca oleracea*, and *Setaria verticillata* [135].

In addition, Dhima et al. [136], determined in the laboratory the phytotoxic potential of *M. verticillata* extracts by the use of a perlite-based bioassay against barnyardgrass (*Echinochloa crus-galli*) and maize (*Zea mays*). They found that mint reduced parameters

like germination, root elongation, and fresh biomass of barnyardgrass in a lower degree compared to extracts from other investigated aromatic plants. The lower significant inhibition of maize germination in the case of mint extracts compared with that of barnyardgrass could be attributed to larger size of maize seeds. In the same study, Dhima et al. [136] investigated the influence of *M. verticillata*, used as incorporated green manure (cover crop) on the presence and development of the following weeds in field conditions: barnyardgrass (*E. crus-galli*), common purslane (*Portulaca oleracea*), puncturevine (*Tribulus terrestris*), and common lambsquarters (*Chenopodium album*), as well as on maize development. The study shown that green manure of mint had a moderate/weak potential for barnyardgrass and some broadleaf weeds control in maize crop.

More recently, Verdeguer et al. [137] evaluated the phytotoxicity of *M. × piperita* oil against the noxious weed *Erigeron bonariensis* (syn: *Conyza bonariensis*) in pre- and post-emergence application in greenhouse environment. The EO of *M. × piperita* showed significant potency at the highest doses (4 and 8 µL/mL), albeit at the lowest rate (2 µL/mL) the germination was even greater than the water control.

The herbicidal effects of *M. longifolia* EO on germination, root and shoot growth of *Rumex crispus* and *Convolvulus arvensis* were studied by Üstüner et al. [63]. They tested four concentrations (5, 10, 15, 20 µg/cm$^2$) of EO, which all inhibited 100% of the evaluated traits in both weeds.

Hanana et al. [55] stated that *M. pulegium* oil almost reduced the germination and seedling growth of *Sinapis arvensis* at 0.5 µL/mL of *Phalaris paradoxa* and *Lolium rigidum* at 0.75 µL/mL. *M. pulegium* oil completely inhibited the same parameters in higher concentrations (0.75 µL/mL for *S. arvensis* and 1 µL/mL for *P. paradoxa*). In that study, the authors declared that the herbicidal action could be related largely to the high amount of oxygenated monoterpenes in the EO.

The herbicidal activity of *Mentha* species (EOs, extracts, etc.) is presented in the Table 4.

**Table 4.** Herbicidal activity of *Mentha* species (EOs, extracts, etc.) against weeds in the cited literature.

| Target Weed | Species of *Mentha* | Tested Compound(s) | Activity/Toxicity | | Method(s)/Doses | Reference |
|---|---|---|---|---|---|---|
| Abutilon theophrasti, Oryza sativa, Datura stramonium, Phalaris paradoxa, | M. spicata | EO [1] | I$_{50}$ [2] of radical length 3.32 2.68 2.70 2.81 | I$_{50}$ of hypocotyl length 3.84 5.92 2.82 2.99 | 1, 2.5, 5, 10 µL/petri | [133] |
| Abutilon theophrasti, Oryza sativa, Datura stramonium, Phalaris paradoxa, | | Carvone | I$_{50}$ of radical length 2.80 2.34 2.17 2.30 | I$_{50}$ of hypocotyl length 3.09 3.02 2.30 2.23 | 1, 2.5, 5, 10 µL/petri | |
| Alcea pallida, Amaranthus retroflexus, Centaurea salsotitialis, Raphanus raphanistrum, Rumex nepalensis, Sinapis arvensis, Sonchus oleraceus, | M. spicata | EO/ Carvone | High inhibitory effect against weed seeds even at low concentrations of EO or carvone | | 3, 6, 10, 20 µL/petri 62.5, 125, 250, 500 µg/mL | [134] |
| Amaranthus retroflexus, Echinochloa crus-galli, Oryza sativa, Portulaca oleracea, Setaria verticillate, | M. spicata | EO | Inhibition of germination | | | [135] |
| Echinochloa crus-galli, | M. verticillata | Extracts | Germination (% of control) 63.7 59.3 | Fresh weight (% of control) 51.5 46.5 | perlite-based bioassay 2 g dry mint/100 mL 4 g dry mint/100 mL | [136] |
| E. crus-galli, Portulaca oleracea, Tribulus terrestris, Chenopodium album, | | Cover crop/ Green manure | Plants/m$^2$ 52 43 47 29 | Fresh Weight g/m$^2$ 62 26 59 29 | In field | |

**Table 4.** *Cont.*

| Target Weed | Species of *Mentha* | Tested Compound(s) | Activity/Toxicity | Method(s)/Doses | Reference |
|---|---|---|---|---|---|
| *Erigeron bonariensis,* | *M. × piperita* | EO | Significant effectiveness at 4 and 8 μL/mL | pre- and post-emergence assays with 2, 4 and 8 μL/mL | [137] |
| *Rumex crispus, Convolvulus arvensis,* | *M. longifolia* | EO | 100% inhibition of seed germination, root and shoot growth | 5, 10, 15, 20 μg/cm$^2$ | [63] |
| *Sinapis arvensis, Phalaris paradoxa, Lolium rigidum* | *M. pulegium* | EO | 100% inhibition of germination and seedling growth | at 0.75 μL/mL at 1 μL/mL | [55] |

[1] EO, Essential Oil; [2] $I_{50}$, Concentration caused 50% inhibition of an evaluated parameter.

According to Dayan et al. [138], 2-phenethyl propionate was a component of *M. × piperita* oil, which is also rich in menthol and menthone. In the same study, it is stated that 2-phenethyl propionate has been patented as an herbicide (like Eco-Exempt[TM] with 21.4% 2-phenethyl propionate or Eco-Smart[TM]), and is a constituent of natural herbicides formulations (like Bioorganic[TM] with 5% 2-phenethyl propionate). The same authors declared that this compound must be diluted before application, and additionally it is absolutely safe to the environment and to human health, since it is a constituent in food flavorings.

Concerning the mode of action, an important mechanism that may explain the control growth of weed species, is the result of allelopathic compounds on the mitochondrial respiration [139]. For this purpose, Mucciarelli et al. [139] investigated whether peppermint EO may affect oxygen uptake in plants using cucumber seedlings. They found that the root and the mitochondrial respiration ($IC_{50}$) were inhibited from the EO and its constituents as following: *M. × piperita* oil by 324 and 593 ppm, (+)-pulegone by 0.08 and 0.12 mM, (−)-menthone by 1.11 and 2.30 mM, and (−)-menthol by 1.85 and 3.80 mM, respectively. Moreover, they concluded that mode of action is associated with the terpenoid interaction with cell walls and plasma membranes. Kombrink and Somssich [140] stated that a series of direct physiological responses via intracellular transduction pathways activates. In addition, terpenoids may interact by causing alterations to the permeability of the plasma membrane and to the fluxe of ions, and possibly diverting oxygen toward an oxidative burst [141]. Studies have shown that the use of menthol caused oxidative stress (via the increase of malondialdehyde levels) [142,143], stomata closure, enhancement of respiration, and swelling of protoplasts [144].

Generally, EOs and their constituents show multi-site activities in plants without high specificity, which is accomplished with synthetic herbicides. In a thorough review by Gitsopoulos et al. [145], a plethora of mechanisms of action were described: inhibition of mitosis, microtubules disruption and cell membrane leakage, inhibitory effects on photosynthesis and decrease of chlorophyll content, disorder of mitochondrial respiration, oxidative stress via the increase of malondialdehyde levels, inhibition of DNA synthesis, stomata closure, enhancement of respiration, and swelling of protoplasts, because of the huge number of EOs components.

*4.2. Crops Phytotoxicity*

*Mentha* EOs or aqueous extracts are likely to present mild to moderate phytotoxicity toward crops like tomato, radish, cotton, or maize, indicating that the *Mentha* species possess good potential to exploit as non-selective bioherbicides in non-crop area, or as selectively applied post-emergence bioherbicides. Furthermore, as far as we know, there is a lack of knowledge concerning the impact of the *Mentha* species or of their main constituents on physiology, growth or yield of other plants in a rotation system, cover crop (green manure), or co-cultivation systems. Finally, many studies have been conducted recently regarding the enhancement of *Mentha* EOs effectiveness in agriculture, and to overcome limitations due to their nature (being unstable when exposed to light and oxygen), by using a combination of EOs, or by developing novel formulations (micro- or nanocapsules).

Mahdavikia and Saharkhiz [146] investigated the stress caused by allelopathic substances of *M.* × *piperita* water extract on germination, leaf area, dry weight, and other physiological and biochemical traits of the tomato. They found that the greatest suppression of the tomato's physiological parameters occurred at a rate of 10% (*v/v*) extract. They also found that several phenolic metabolites, such as ellagic acid, hesperidin, sinapic acid, and trans-ferulic acid were identified in *Mentha*'s aqueous extracts, giving a possible explanation for the inhibition of germination and seedling growth of tomato due to an induced oxidative stress. Similarly, the treatment of tomato seeds with the EO of *M. suaveolens* at MIC (0.78 mg/mL against *M. michiganensis* and *Pseudomonas savastanoi* pv. *savastanoi*) and 4 × MIC using petri plate assays inhibited more than 60% the tomato seeds germination [33].

Moreover, Mahdavikia et al. [147] demonstrated that substances in the water extract of *M.* × *piperita* had considerable influence on radish (*Raphanus sativus*) growth, specifically in total soluble sugars, biochemical compounds such as proline and phenols, membrane permeability, and antioxidant enzymes.

With regard to sustainable vegetable production, Ulbrich et al. [148] studied the greenhouse cultivation of white cabbage supplemented with two *M.* × *piperita* varieties. They concluded that, when applied to the young sensitive stages of the *Brassica* seedlings, *Mentha* volatiles enhanced the productivity and increased the quality and quantity of the aboveground biomass. The same researchers mentioned no promoting effect on leaf development or leaf weight was found when white cabbage seedlings were exposed to menthone, menthol, or their 1:1 combination.

Skrzypek et al. [149] studied the effectiveness of water extracts on *M.* × *piperita* leaves at doses ranging from 1 to 15% on germination and on the physiological parameters of sunflower (*Helianthus annuus*) grown in greenhouse conditions for 30 days. They found that increasing the concentrations of peppermint aqueous extracts caused a damage to photosynthesis (a reduction of chlorophyll a and a gain of chlorophyll b content), a deleterious effect on germination, and an increase of electrolytes leakage in cell membrane of sunflower seedlings.

Recently, Synowiec et al. [150] found that microencapsulated *M.* × *piperita* oil caused phytotoxicity on maize, even at the minimum dose (36 g/m$^2$), as evidenced by the obstruction of maize emergence, and by the decrease in both growth and biomass accumulation. In addition, Karkanis et al. [151] demonstrated that the inclusion of *M.* × *piperita* and *M. spicata* in a crop rotation system with maize provoked a deleterious effects on the growth and grain yield of maize, probably due to the allelopathic action of both *Mentha* species. In another study, Synowiec et al. [152] determined the effect of soil-maltodextrin microencapsulated EO of *M.* × *piperita* (12%) at tested doses of 0.75, 1.5, and 3 g per pot on young seedlings of maize, as well as on *Echinochloa crus-galli* and *Chenopodium album* weeds in a pot experiment. Their results showed that maize was the most resistant to the microencapsulated *M.* × *piperita* oil, although reductions were observed on chlorophyll content, whereas *C. album* was more susceptible to the microcapsules than *E. crus-galli*.

The phytotoxicity of iso-menthone, pulegone, carvone, piperitone, piperitone oxide, and piperitenone oxide, which are constituents of *M. spicata* and *M. pulegium* EOs, have been studied in lettuce, tomato, and ryegrass [87]. The results demonstrated that the aforementioned compounds at microplate well concentrations of 0.4 and 0.2 mg/mL exhibited phytotoxic activity, as germination and the leaf and root growth of tested plants were considerably inhibited.

Moreover, innovative approaches are proposed to exploit in horticulture *M. spicata*, as dry raw material, incorporated into the culture substrate, in order to rapidly produce robust tomato seedlings [153]. Similarly, Chalkos et al. [154] reported the considerable enhancement of tomato growth and, simultaneously, the inhibition of weed emergence caused by the incorporation of *M. spicata* compost in growth media at a dose of 2 to 8% *w/w*). Similarly, it was demonstrated that soil amendment with 4% (*w/w*) *M. spicata* dried plant material improved the tomato tolerance against *Fusarium* and *Verticillium* wilts, soil fertility,

and subsequently increased the yield and product quality in greenhouses [45]. However, according to Karkanis et al. [151], the introduction of *M. × piperita* and *M. spicata* in a three year crop rotation system with maize adversely influenced the maize crop, and this could probably be related to the allelopathic potential of both *Mentha* species. Consequently, it is important to consider the deleterious effects on the successive crops, notwithstanding the potential benefits of such a crop rotation system, concerning weed control and the high added value of the final product derived by *Mentha* cultivation [151].

Another feature of *Mentha* species reviewed by Gholamipourfard et al. [52] is the phytoremediation aspect of *Mentha × piperita*, which has the ability to accumulate high concentrations of heavy metals-ions like Cr, chromate and Cu from soil.

Taking into consideration all the abovementioned research, we conclude that the *Mentha* species could be used in sustainable agricultural systems for integrated pest management. This can be achieved by utilizing them either directly as crop, green manure, or compost, or for the development of natural pesticides consisting of *Mentha* EOs or extracts. Nevertheless, further field experiments must be conducted to confirm the efficacy of various formulas on pests under crop conditions.

## 5. Modes of Application

EOs may be utilized as biopesticides by different application methods, such as direct contact with pests, ingestion, and inhalation (by fumigation), which is the most common method [116]. Spraying of EOs or adding EO traps have been also used. It should be noted that the efficacy of each method should be evaluated case by case, and results from laboratory assessments may be different when they are tested in field trials [25]. The regime, time of exposure, and application method are the parameters influencing the effect of a biocide. The time and dose should be low enough to avoid harmful effects on other organisms [25]. Concerning the use of *M. pulegium* oil as a pesticide, its potential toxicity should be taken into consideration, due to its constituent pulegone. In Europe, there are limitations on the addition of pulegone in foods, and its use as flavoring is banned [155], though in the USA, *M. pulegium* oil permitted as food additive, and flavoring [156]. *M. pulegium* EO, in high doses is toxic to humans, having fatal consequences, such as hepatotoxicity and cardiovascular failure [25].

## 6. Constraints and Perspectives

Due to the widespread misuse of synthetic pesticides resulting in the development of pesticide-resistant pests, and increasing consumer awareness concerning their negative impact on public health and the environment, the application of biopesticides as pest control tools in sustainable/organic production is of major importance.

Plant based products and particularly EOs, known for their antimicrobial and herbicidal activities, could be successfully used as useful tool in the sustainable and organic farming, by utilizing them as biopesticides, as an alternative to synthetic ones. However, the disadvantage of not dissolving in water, their sensitivity to light and oxygen, and their high volatility, are some constraints concerning their wide application in agriculture [60]. On the other hand, the majority of them are considered GRAS (Generally Considered as Safe), biodegradable, having low toxicity in mammals and the environment, which are advantages for their exploitation as natural pesticides, finding at the same time solutions for effective formulations, to overcome these restrictions.

Different techniques have been developed, i.e., the incorporation of EOs into films and particles to decrease the diffusion of EOs, thus achieving controlled release to the applied surface over the time. Among them, encapsulation in different matrices is a successful technique for protecting bioactive compounds like EOs from degradation, evaporation, harmful environmental issues, or mechanical stress. Remarkable studies have been carried out in recent years on EOs encapsulation [25]. Kavetsou et al. [157] studied a method for the encapsulation of *M. pulegium* EO in *Saccharomyces cerevisiae* microcarriers, and the insecticidal action towards *Myzus persicae*. They confirmed the efficacy of the EO

and estimated that encapsulation enhanced the action of the EO against the insect by three days. Encapsulation of *M. × piperita* EOs in different matrices such as maltodextrin, modified starch, gelatin/gum arabic has been studied, whilst menthol was encapsulated in cyclodextrin successfully, improving its physicochemical properties, and sustaining its release [60]. Additionally, the encapsulation of *M. × piperita* EOs in chitosan–cinnamic acid nanogel promoted the EO antifungal effectiveness on *Aspergillus flavus* [46].

Moreover, for the industrial production and commercialization of EOs as biopesticides, the different constitution of *Mentha* oils and the various chemotypes within the same species, reflecting to different ratios of the bioactive compounds, should be taken into consideration. Therefore, standardized procedures and EOs/extracts should be used, to achieve the same biological effect and consistency [25]. In the case of menthol mints' EOs, it was demonstrated that the antimicrobial activity is associated with menthol content, which is more effective than menthone [60]. Additionally, the combination of EOs with other means, or using different mixtures of EOs, is a common and efficient strategy to control pests.

Although *Mentha* oil is already used in commercial products as biocide [14], and menthol is already the ingredient of numerous industrial products, several issues concerning residual phytotoxicity, long term effects on the environment, and biodiversity, i.e., on non-target microorganisms such as pollinating insects and natural predators, should be assessed. Natural substances should be properly and scientifically confirmed for repeated application usage before their approval [14], whilst standardization, toxicity, and regulatory issues should be considered before commercialization.

Concerning the use of *Mentha* species in crop rotation system, the high added value of *Mentha* cultivation and the total economic benefit, along with a possible negative allelopathic activity with the cultivated crops, should also be taken into account [151].

## 7. Conclusions

Plants of the genus *Mentha* and their products such as EOs and extracts, could be used as alternative biopesticides to synthetic ones due to their bioactivity. However, future research should be focused on efficient and cost-effective formulation methods, the identification of the bioactive compounds associated with the specific bioactivity, while long term studies regarding their impact on the environment and biodiversity should be carried out. Additionally, there is a need for more field experiments in order to scale up and validate the laboratory results, and particularly to estimate the duration of protection in the field/or greenhouse, and the effectiveness of *Mentha* EOs' products as commercial biopesticides. To this end, the study of other *Mentha* species that have not yet been commercialized, such as wild growing, species native to Greece (i.e., *Mentha longifolia* subsp. *petiolata* or *M. × villoso-nervata*), may reveal substantial bioactivity and stong prospects for their potential use as biopesticides.

Based on the reported properties and the related biological activities affecting the pre- or post-harvest plant pathogens, animals, weeds, and crops, *Mentha* species present further challenges for their utilization in sustainable agriculture. Thus, based on the knowledge gained so far and with the ai of promoting the use of biopesticides—and also in alignment with effective regulations—we consider that the use of *Mentha* species and their products (in all the above forms) could be capitalized on as part of an integrated pest management system.

**Author Contributions:** Writing–original draft preparation, K.I.K.; Writing–review and editing, P.C. and K.I.K. All authors have read and agreed to the published version of the manuscript.

**Funding:** This research received no external funding.

**Institutional Review Board Statement:** Not applicable.

**Informed Consent Statement:** Not applicable.

**Data Availability Statement:** Not applicable.

**Conflicts of Interest:** The authors declare no conflict of interest.

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
