# Peer review of "Approaches and Applications of Mentha Species in Sustainable Agriculture"

_sustainability, doi:10.3390/su15065245_

Round 1
Reviewer 1 Report
Although the study Approaches and Applications of Mentha species in Sustainable Agriculture has good potential in being interesting and there is a lot of work/literature research, it must be completed and corrected in some points. In this regard, I suggest the following:
L31. “is very important” = a statement saying nothing. All species of plants are “very important”. Please reshape.
Please check the Instructions for authors regarding reference insertion in the text, MDPI style, in brackets, and apply. Instructions for authors are given to be respected, they are not optionally.
L113-116. Should be removed, as what is presented in each section, subsections, paragraphs is obvious.
Table 1. ml must be written as mL, as Litter is the unit of measure for volume in the International System. Please revise/correct in the entire manuscript and be consistent with denotation as in some places you have used correctly “L” for litter and its submultiples. Also, extend all the tables on the entire width of the page, as the MDPI draft allows it, and the tables will look much better and not so crowded.
Abbreviation EO must be also explained under the table. Check the Instructions for authors regarding abbreviation.
L233-234 must be multiple referenced as it states that “researchers have shown a strong interest”. I suggest checking and referring to https://doi.org/10.3390/antiox11071359 , https://doi.org/10.1016/j.biopha.2022.113161 and https://doi.org/10.37358/RC.20.1.7854
Tables 2 and 3. μl must be written as μL, and μl/l as μL/L, as I explained above.
References must be in the order they appear in the text, not in alphabetical order. Please check the Instructions for authors regarding reference insertion in the text, MDPI style. Proceed consequently
Reviewer 2 Report
Minor comment:
Line 100: delete the typo 'are due'
General comment:
This is an excellent study, well covered by self-explainable tables. However, the text is too long and sometimes narrative, in detail explaining the previous studies. Some parts could be much shorter. In its current form, it does provide all essential information in one place, but it is too long.
The introduction is well-written, sub-headings are appropriately divided, containers and perspectives are elaborated and references are well chosen.
Reviewer 3 Report
The authors have compiled an interesting “Approaches and Applications of Mentha species in Sustainable Agriculture” article with comprehensive information. However, they need to improve the quality of the presentation of the tables and the reference writing as per journals guideline.
